# Experimental Testing on Tuned Liquid Dampers for Implementation in Industrial Chimneys

**DOI:** 10.3390/s24092800

**Published:** 2024-04-27

**Authors:** Giancarlo Marulli, Carlos Moutinho

**Affiliations:** CONSTRUCT, ViBest, Faculty of Engineering (FEUP), University of Porto, R. Dr. Roberto Frias S/N, 4200-465 Porto, Portugal; moutinho@fe.up.pt

**Keywords:** vibration control, TLD, passive damping, shaking table tests, customised sensors

## Abstract

A TLD is a passive damping device that works by dissipating energy through the sloshing of the liquid and the effect of wave breaking, thereby controlling the vibrations of the structure. One of the applications where TLDs are of great interest is in the case of industrial chimneys since these structures often have a very low natural frequency, which can be easily achieved in a control device of this type. The main objective of this study is to evaluate the behaviour of an annular TLD composed of multiple cells through laboratory tests and investigate if it is adequate to design it as an agglomeration of smaller rectangular TLDs. The influence of the amplitude of displacement on the behaviour of the annular TLD will also be analysed. The tests were performed on a shaking table and recurring with pendulums of the same length but of different masses. Three reservoirs were studied as TLDs: a rectangular one, a cell of an annular TLD and a quarter-ring of an annular TLD. This study concluded that the analytical methods developed in previous studies were, in general, adequate for the design of a rectangular TLD and that it was reasonable to design the annular TLD studied as a combination of rectangular ones, as its cells were a close match to a rectangle of similar dimensions. It was also concluded that a compartmentalised annular TLD is an adequate solution for the vibration control of structures with high displacements.

## 1. Introduction

Chimneys have been used to conduct and disperse fumes and other undesirable gases since Antiquity. Being tall and slender structures with low damping coefficients, they are vulnerable to dynamic forces mainly caused by the wind. The wind is particularly prejudicial to chimneys due to the formation of a vortex-shedding effect caused by the wind breaking when flowing around the chimney, which causes the structure to vibrate [1,2,3].

As is well established, dynamic actions (such as earthquakes or the wind) can induce vibrations in a structure that may lead to degradation, loss of stability and even failure. Even when the level of vibration detected in the structure is not severe enough to put its integrity into jeopardy, it may lead to discomfort on the part of the occupants of the building [3,4,5,6,7,8].

When this happens, the introduction of vibration control systems with the intent to reduce the level of vibration is justified to prevent structural damage or to increase the level of comfort for the occupants [3,4,9].

The TLD should be implemented at the point of maximum displacement of the most critical mode of vibration. In the case of chimneys, the most critical mode of vibration is the first one, with maximum displacement at the top of the structure [6,8,10]. The amplitude of vibration of the first mode should not exceed the indicative value of acceptable vibrations of 5% of the external diameter of the chimney when assessing the vortex-shedding amplitude of vibration of the stack [11].

The devices used to control vibrations in this study were tuned liquid dampers (TLDs), a kind of passive damping device initially developed for utilisation in Aerospace and Marine Engineering at the beginning of the 20th century [12]. Solutions utilising TLDs have become popular due to their ease of implementation, simplicity and small investment and maintenance requirements when compared to other damping systems. It is also a kind of solution that has been used for suppressing wind-induced vibrations of tall structures, such as chimneys [12,13,14].

The basic principles behind a tuned liquid damper (TLD) are similar to that of a tuned mass damper (TMD) [12]. A secondary mass (in this case, a body of liquid) is introduced into the structural system and tuned to act as a dynamic vibration absorber. However, in the case of TLD, the damper response is highly nonlinear due either to liquid sloshing or the presence of orifices [13,15,16].

The major difference between a TMD and a TLD is that, due to the complex interaction between the liquid and the reservoir walls, a simple single degree of freedom (1-DOF) analogy is an improper simplification in the latter case. Nevertheless, TLDs have a near-zero acceleration trigger level, which makes them very attractive for damping horizontal structural sway with very low frequencies [7,14,17,18].

The behaviour of the liquid inside the reservoir depends on many factors, varying from the height of the liquid inside it, its shape, its amplitude of displacement, etc. [6,10,18,19,20,21,22]. This will impact the linearity of the liquid as well as the behaviour of the waves on its surface (Figure 1 and Figure 2).

### 1.1. Sloshing Frequency

The formula for the liquid sloshing frequency of its first mode of vibration (therefore for the sloshing frequency of the TLD), shown as Equation (1), is derived from the linear wave theory, developed by Lamb in 1932, and can be reliably used in the design of a rectangular TLD [23,24,25,26].
(1)$$f_{f}=\frac{1}{2\times \pi }\times \sqrt{\frac{\pi \times g}{L}\times tanh(\frac{\pi \times h_{f}}{L})},$$
with $f_{f}$ being the sloshing frequency of the TLD [Hz], g being the acceleration of gravity [m/s^2^], L being the length of the TLD [m], and $h_{f}$ being the height of water [m].

Equation (1) is not precise, which leads to a small error in the estimation [18,23]. An equation to correct this error in sloshing frequency in small amplitude ranges was developed [18], but this correction will not be used because the estimation error observed in this study was not significant enough to warrant its implementation.

### 1.2. Ratio of Masses and Liquid Depth Ratio

The ratio of masses and liquid depth ratio are important parameters for the design of a TLD. The first one is obtained by comparing the mass of the liquid inside the TLD with the mass of the structure as per Equation (2), with its ideal value in the interval from 5% to 10% [20]. The second one is obtained by comparing the height of the liquid with the length of the reservoir as per Equation (3), with its ideal value being inferior to 0.20 and preferably close to 0.10 [23].
(2)$$\mu =\frac{M_{TLD}}{M_{str}},$$
with μ being the ratio of masses, $M_{TLD}$ being the mass of the liquid [kg], and $M_{str}$ being the mass of the structure [kg].
(3)$$\varepsilon =\frac{h_{f}}{L/2},$$
with ε being the liquid depth ratio.

### 1.3. Effects of Multiple Rectangular TLD on a Structure

The effects of multiple TLDs (MTLDs) on a structure started being studied in the early 1990s [23,27]. For the purpose of this study, the main parameters of interest will be the central frequency and the central liquid depth ratio, Equations (4) and (5), respectively.
(4)$$f_{0}=\frac{f_{highest}+f_{lowest}}{2},$$
with $f_{0}$ being the central frequency [Hz], $f_{highest}$ being the sloshing frequency of the TLD with the highest natural frequency [Hz], and $f_{lowest}$ being the sloshing frequency of the TLD with the lowest natural frequency [Hz].
(5)$$\varepsilon_{0}=\frac{\varepsilon_{highest}+\varepsilon_{lowest}}{2},$$
with $\varepsilon_{0}$ being the liquid depth ratio, $\varepsilon_{highest}$ being the liquid depth ratio of the TLD with the highest being the liquid depth ratio, and $\varepsilon_{lowest}$ being the liquid depth ratio of the TLD with the lowest being the liquid depth ratio.

These parameters are analogous to the sloshing frequency and the liquid depth ratio, as indicated by their names. Equation (2) is still valid for the determination of the ratio of masses of an MTLD system.

### 1.4. Damping

The decay in free vibration is the most frequently used method of finding the viscous damping ratio ξ of a system through experimental measurements. The damping ratio can easily be determined from the ratio of two peak displacements measured over n consecutive cycles (considering that the system has been set into free vibration) [28].

Equation (6) can be used to estimate the damping coefficient of a system in free vibration.
(6)$$\xi =ln(\frac{u_{1}}{u_{n+1}})/(2\times n\times \pi ),$$
with ξ being the damping coefficient, $u_{1}$ being the displacement value at the start of the free vibration, $u_{n+1}$ being the displacement value after n cycles of the free vibration, and n being the number of cycles of the free vibration.

The effect of damping of liquid motion is significant near resonance and hence must be taken into account in the modelling of the tuned liquid damper. This can be achieved with Equation (7).
(7)$$\lambda =\frac{1}{\left(\eta +h_{f}\right)}\times \frac{1}{\sqrt{2}}\times \sqrt{\omega \times \nu }\times (1+\frac{2\times h_{f}}{b}+S),$$
with λ being a constant characterising the damping of liquid motion, which accounts for the effect of the bottom, side wall and free surface [rad/s]; η being the maximum liquid surface elevation [m]; ω being the excitation frequency [rad/s]; ν being the kinematics viscosity of the liquid [m^2^/s]; b being the width of the reservoir [m]; and S being a “surface contamination” factor that accounts for damping due to stretching effect in the contaminated liquid surface.

Following the steps of previous studies, a unitary value was taken for the surface contamination factor [23,24,29,30].

Equation (7) does not take into consideration the effect of wave breaking when determining the damping of liquid motion. A coefficient can be introduced to account for this effect in the damping of the TLD [23]. This coefficient should not be relevant for the present study, however, as the conditions of the tests were designed to avoid the wave breaking in effect.

The damping ratio of the TLD can be determined from the previously mentioned constant using Equation (8).
(8)$$\xi_{TLD}=\frac{\lambda }{2\times \omega_{f}},$$
with $\xi_{TLD}$ being the damping ratio of the TLD, and $\omega_{f}$ being the sloshing frequency [rad/s].

### 1.5. Pendulum Mechanics

As the laboratory tests were envisioned to be carried out with the use of a shaking table, it was decided that a pendulum would be the best way of constructing a 1-DOF system possessing similar characteristics to that of an industrial chimney on its critical mode of vibration, namely their low damping coefficient, high displacement (even for small excitation) and the possibility of working at low frequencies. Using a pendulum also made it possible to avoid some interferences, such as the friction between moving components, which could have a big impact on the results.

The frequency of the pendulum depends only on its length, as can be seen in Equation (9).
(9)$$f=\frac{1}{2\times \pi }\times \sqrt{\frac{g}{L_{p}}},$$
with f being the frequency of the pendulum [Hz], g being the acceleration of gravity [m/s^2^], and $L_{p}$ being the length of the pendulum [m].

### 1.6. Objectives

The main objective of this study is to evaluate the behaviour of an annular TLD composed of multiple cells through laboratory tests, if it is adequate to design it as an agglomeration of smaller rectangular TLD disposed around a circumference with different angles to the acting excitation and its applicability in the control of vibrations in industrial chimneys. Annular TLDs were chosen as the focus of this study because reservoirs with this shape are geometrically compatible with industrial chimneys, making it easy to implement them on the structure. In addition, they can control horizontal vibrations that occur in more than one direction, such as those caused by the change in the winds.

As a necessity for the main goal, tests will be made to evaluate the current analytical formulas used in the design of rectangular TLDs, focusing on the influence of the liquid height and reservoir length on the performance of the TLD.

The influence of the input direction on the performance of a rectangular TLD will also be evaluated before finally transitioning from the rectangular reservoir to an annular TLD composed of several quasi-rectangular cells, at which point the knowledge accumulated in the previous tests will be applied to further this study’s main goal.

This study is of particular relevance as experimental tests with non-rectangular TLDs are still less common than those with rectangular ones, especially tests with TLDs of this particular configuration [5]. Additionally, customised sensors developed in the context of the activities of the research group were also used, which facilitated the measurement of the various variables involved in the studied system.

## 2. Materials and Methods

### 2.1. Shaking Table

The shaking table used in this study was designed at the University of Porto. It has a surface area of 3 × 3 m^2^ and has a maximum amplitude of 440 mm in both horizontal axes, and also has movement capability on its third axis (vertical). The table can reach frequencies of up to 30 Hz and a maximum speed of 670 mm/s with a maximum load of 10 tons.

Despite the capabilities of the shaking table, it is well adjusted to perform tests with very low amplitudes and frequencies as well, as is the case in this study. For the study in question, only one of the main axes will be in use, and its amplitude was conditioned to a maximum of 5 mm.

In this study, only one axis of the horizontal axis of the shaking table was utilised (specifically, the first axis, which was aligned with the pendulums), producing harmonic excitation with an amplitude of 0.5 mm (that will be used in all tests). This amplitude was chosen because it caused the displacements of the pendulums to be in the range of displacements of the chimneys for which the annular TLD was designed (which ranged from 2 cm to 5 cm). It should be mentioned that tests were also performed with an amplitude of 0.25 mm, but as the results were very similar to those with the former amplitude, they were not included in this study.

### 2.2. Pendulum Systems

Two pendulums were used in this study. A smaller one is made out of wood and uses metal plates to increase its mass, and a larger one is made out of steel and uses a wooden structure and a concrete slab to increase its mass. In both cases, the length of the pendulum (i.e., the free length of the cables connecting the fixed part of the structures with their mobile ones) was set to 505 mm so that its oscillatory frequency lies in the frequency range of the most critical mode of vibration of the chimneys for which the annular TLD was designed (which ranged from 0.65 Hz to 0.78 Hz).

Applying the length of the pendulum to Equation (9), the resulting frequency is
f=0.698 Hz.

Regarding the smaller pendulum system, a structure made out of wooden rafters was built to serve as the support of the pendulum, which was made out of four steel cables connected to a wooden box designed to hold the TLD reservoir and additional masses. The cables were braced to impose a single axis of movement in the pendulum.

The wooden structure was fixed to the vibrating table using screws and clamps. The pendulum’s box was made out of the same plates and rafters as the rest of the woodwork, with a usable area of 785 × 630 mm^2^ (Figure 3 and Figure 4).

The mass to be considered for this structure has four components: the mass of the pendulum (the wooden box and the cables), which was about 9.5 kg; an additional mass comprised of four 600 × 300 × 20 mm^3^ steel plates, each of which weighed approximately 28.08 kg; an undetermined number of 600 × 150 × 10 mm^3^ steel plates, each of which weighed approximately 7.15 kg; and the weight of the reservoir used in the tests when empty. The number of the latter steel plates will depend on the height of water in the reservoir and will be defined in Section 3.

The larger pendulum, made out of four metallic columns, was built to serve as the support of a structure made out of four steel cables connected to two transverse metallic beams, which were connected longitudinally by three steel rebars. As before, the cables were braced to impose a single axis of movement in the pendulum.

The pendulum held a wooden structure weighing approximately 35 kg, which supports a concrete slab of approximate surface area of 2 × 2 m^2^ and weighing about 816 kg. The weight of the reservoir used in the test when empty will also be added when determining the mass of the structure. The larger pendulum can be seen in Figure 5 and Figure 6.

### 2.3. Reservoirs

Three reservoirs were used in this study: a rectangular reservoir and two reservoirs based on a compartmentalised annular TLD, one of which was built to match a single cell of the annular TLD and the other was built to match a quarter of the annular TLD. The first two reservoirs were tested on the smaller pendulum and the other one on the larger pendulum.

The rectangular reservoir was made of 10 mm thick transparent acrylic plates with internal dimensions of 600 × 400 × 400 mm^3^ and weighing about 12.25 kg (Figure 7 and Figure 8).

The annular TLD was designed based on a real prototype that will be installed on an industrial steel chimney to control vibrations due to wind forces (Figure 9).

This prototype was designed to be built with steel cells and has an external diameter of 3.932 m and an internal diameter of 3.080 m (therefore, each cell has a length of 426 mm). Its height is 492 mm, and the divisions are made with thin steel sheets, creating 20 cells of equal dimensions; the divisions have a height of 458 mm and two small triangular punctures in their base, allowing the liquid to flow between the cells.

The TLD composed of a single-ring cell was built with the same dimensions as one of the cells of the annular reservoir described before. Its bottom and rectilinear walls were made of 15 cm thick plywood rigid panels, and the curved walls were made of 4 mm thick plywood flexible panels. The interior surfaces of the reservoir were isolated with white silicone paint to prevent leaking (Figure 10 and Figure 11).

The quarter-ring reservoir was built following the same design as the real annular TLD, in this case with 2 mm thick welded steel sheets and weighing about 78 kg (Figure 12 and Figure 13). The quarter ring TLD ends up being composed of five cells.

### 2.4. Sensors

For the smaller pendulum, displacements were measured using a Sharp sensor, while, for the bigger one, an accelerometer was used.

A water level sensor was also in use to measure the liquid height inside TLDs, which is composed of a Milone Technologies’ (Sewell, NJ, USA) eTape™ Continuous Fluid Level Sensor PN-12110215TC-12 (Figure 14).

The eTape sensor is a continuous fluid level sensor for measuring levels in water and other non-corrosive liquids. As the sensor’s envelope is compressed by hydrostatic pressure of the fluid in which it is immersed, there is a change in resistance with a resistive output that is inversely proportional to the level of the liquid—the lower the liquid level, the higher the output resistance and vice versa [31].

This sensor was manually calibrated using a linear regression curve, which was close to the typical output characteristics provided by the manufacturer.

The eTape was submerged in the liquid and fixed to the wall of the TLD with a 3D-printed support designed to comply with the best working conditions presented by the datasheet while still allowing the sensor to measure the surface elevation of the liquid in real time.

The displacement sensor used to measure the TLDs’ reservoir motion was a Sharp Corporation’ (Sakai, Osaka, Japan) GP2Y0A51SK0F line sensor with a range of 2 to 15 cm (Figure 15).

GP2Y0A51SK0F is a distance-measuring sensor unit built from a position-sensitive detector (PSD), an infrared light-emitting diode (IR-LED) and signal processing circuit. By adopting a triangulation method, the reflectivity of the object, the environmental temperature and the operating duration have little influence on the distance detection [32].

The sensor output is a voltage that corresponds to the detected distance (becoming lower when the distance increases), which means that this sensor can easily be used as a displacement sensor. It should also be pointed out that the resolution of the sensor is higher when working in a shorter range.

The sensor was manually calibrated using a polynomial regression curve, which has the typical output characteristics provided by the manufacturer.

The Sharp sensor was positioned and fixed to the shaking table with a 3D-printed support designed to keep the sensor at the same level as the pendulum and for its position to be easily adjustable in order to keep the sensor within its higher resolution range.

The accelerometer used to measure quarter-ring TLD motion was an Analog Devices’ (Wilmington, MA, USA) ADXL355 sensor (Figure 16). The reason why an accelerometer was used instead of a displacement sensor is because the motion amplitude may be very large in this case. Due to the fact that the system response is practically harmonic, the integration of acceleration to displacement is a simple and rigorous process.

The ADXL355 is a digital output 3-axis accelerometer with selectable measurement range, low noise density, low gravity offset drift and low power consumption. It supports a ±2.048 g, ±4.096 g or ±8.192 g range. It also offers a minimal offset drift over temperature and long-term stability, enabling precision applications with minimal calibration [33].

This accelerometer has an internal antialiasing filter, which was configured to work on a data acquisition frequency of 62.5 Hz. Its output was the measurement in all axes, although only the excited axis (zz with the device as the reference point) was analysed. The data were collected in counts and then converted to displacement.

The accelerometer was fixed on the larger pendulum with a magnetic support.

While in the case of the accelerometer, the data were recorded locally on the module, in the case of the water level and Sharp sensors, the data were acquired using National Instruments’ (Austin, TX, USA) NI-9219 and NI-9215 universal analogue input modules connected to a NI cDAQ-9178 compactDAQ chassis integrated with National Instruments’ LabVIEW software (version 2019).

## 3. Results and Discussion

### 3.1. Tests with the Rectangular Reservoir Not Instaled on the Pendulum

The initial tests were performed with the rectangular reservoir set directly on the shaking table with the goal of evaluating the applicability of the formulas for sloshing frequency and damping of liquid motion in the current study and also to determine how the water would behave when the reservoir was subjected to an excitation that was not parallel to one of its main axes (when the reservoir was rotated relatively to the excitation direction). Table 1 contains the conditions in which these tests were performed.

For all configurations, frequency tests were performed in a predetermined frequency range through the use of discrete values (0.02 Hz apart or 0.01 Hz apart while near the resonant frequency). Each test consisted of the excitation of the reservoir for 10 min in order to achieve the steady-state response of the liquid.

For P0 and P90, the tests were performed with a frequency range close to the expected resonant frequency (0.70 to 0.80 Hz and 1.00 to 1.10 Hz, respectively). For the other cases, the tests were made in both of these ranges with the intention to cover the intermediary frequencies (0.80 to 1.00 Hz) if necessary.

The measurements of the variation of the liquid height were recorded for all the tested frequencies in all the different configurations, both when the excitation was present and also in free decay vibration.

The steady-state response of the liquid in the different frequencies for each configuration was then plotted, creating frequency response graphs, in order for the resonating frequency of the liquid to become evident in each case.

The behaviour of the reservoir in its main axes was close to what was expected, with the maximum surface elevation in a steady state occurring when the frequency was 0.74 Hz for P0 and 1.08 Hz for P90. For P30, P45 and P60, it was observed that no matter the angle the excitation made with the main axes of the reservoir, the liquid behaved as if it was being excited in those main axes according to the frequency of excitation, i.e., when the frequency was close to 0.74 Hz, the liquid behaved similarly to its behaviour in P0 and, when the frequency was close to 1.08 Hz, to its behaviour in P90. This resulted in two frequencies appearing as a resonance behaviour (Figure 17 and Figure 18).

The damping of liquid motion for the resonant frequency was calculated using Equation (7), with the liquid surface elevation measured in the steady-state phase of the response. For P30, P45 and P60, the width of the reservoir was assumed based on the behaviour of the water because of the aforementioned similarity of the liquid behaviour and the movement of the water in their main axes.

A summary of the information gathered in these tests is presented in Table 2.

As can be seen, configuration P30 had a symmetrical behaviour between its lower resonance response and its higher one, but with the increasing of the rotation in configurations P45 and P60, the response in the higher resonance became more prevalent. The damping ratio of the TLD was very low, no matter the configuration, despite the high surface elevation reached. This was expected due to water being the liquid used.

### 3.2. Tests with the Reservoirs on the Smaller Pendulum

In the first stage, tests with the smaller pendulum without TLD were performed to confirm its natural frequency and to determine its response with no vibration control. Figure 19 depicts the frequency response curve of these tests based on the steady-state response of the pendulum to a base excitation amplitude of 0.5 mm.

As can be seen, the resonant frequency was 0.69 Hz, which is very close to what was previously calculated. The damping ratio of the structure was determined through the free vibration decay expressed analytically in Equation (10).
(10)$$u(t)=0.0278\times e^{-0.0162\times t},$$
with u(t) being the displacement amplitude [m] and t being the time [s].

Recurring to Equation (6), it is possible to determine the damping ratio of the pendulum, resulting in
ξ=0.37%

A summary of the information gathered in these tests is presented in Table 3.

The maximum displacement of the pendulum is relatively high because the damping ratio, as expected, was very small.

#### 3.2.1. Interaction of the Rectangular TLD with the Smaller Pendulum

Due to preliminary studies, it was determined that the best position to be assumed by the TLD would be with its longer side perpendicular to the displacement of the pendulum [20].

In this configuration (denoted as R0; shown in Figure 20), the length of the TLD is 0.4 m. Returning to Equation (1) to determine the height of water to be used in order to tune it to the structure, the result is
$$h_{f}=3.25\;cm$$

With this information, the ratio of masses and the liquid depth ratio can be determined as per Equations (2) and (3), respectively:μ=4.41%   ε=0.16

As it was expected that the structural displacement would be reduced, the proximity sensor was relocated to a position closer to the structure to guarantee a greater resolution in the collected data. Table 4 contains the conditions in which these tests were performed.

Figure 21 shows the magnitude of the steady-state response in all rotating following, and Table 5 summarises the results obtained.

The attenuation of the structure response in most cases was close to 80%, which is equivalent to an increase in the structural damping coefficient by a factor of five.

The results are in line with what was expected from the initial assumptions, with the TLD performing better when its main axis was aligned with the movement of the structure (configuration R0) with a decrease in the control of the structural displacement whenever the rotation of the TLD increases.

It should be noted that in the lower frequencies (0.60 to 0.75 Hz), it was clear that two waves were being formed on the liquid and, specifically at lower resonance, the noise reached a point in which the waves were close to the breaking point, and a wave in the other main axis was also in effect. It was also noted that starting at frequency 0.82 Hz, the water movement left the main axes, adopting a diagonal movement in between both axes, although a complete axis shift did not occur in the measured interval. This nonlinear behaviour is common when dealing with TLD, as previously stated.

#### 3.2.2. Interaction of the Ring Cell TLD with the Smaller Pendulum

When the cell was aligned with the movement of the pendulum (denoted as configuration C0; shown in Figure 22), the length of the TLD was 0.426 m. Using Equation (1) to determine the height of water to be used in order to tune it to the structure, the result is
$$h_{f}=3.7\;cm$$

Due to the different masses of the reservoirs, to keep the structural mass as close as possible, an additional 7.15 kg metal plate was put on the pendulum. With this information, the ratio of masses and the liquid depth ratio can be determined as per Equations (2) and (3), respectively:μ=4.85%   ε=0.17

Six configurations were studied for this reservoir in order to fully cover all positions a cell could assume inside the annular TLD (Table 6).

Visual observation of the sloshing of the water inside the ring cell reservoir showed that it flowed radially for most configurations (as if it had started on the centre point of the ring) with no major disturbance until the resonance was reached. For the lower frequencies, the formation of two waves was noticed, as was the case for the previous tests, and no axis shift occurred. For configuration C90 only, the water flowed along the reservoir’s curved sides (i.e., aligned with the movement of the pendulum). There was no noticeable wave breaking.

Given the assumption that the ring cell will behave like a rectangular reservoir, configuration C0 should perform similarly to configuration C90 due to the similarities between both tests. This seemed to be the case.

Configuration R0 (with the rectangular reservoir) performed better at the lower end of the studied frequency interval, but both performed similarly to one another at the higher end. Configuration C0 provided a displacement attenuation of 71% compared to R0’s 82%. The displacement peak caused by the beating was slightly higher for C0 as well. That being said, the ring cell TLD still had a good performance, although a slight positive mistune might be in effect.

In order to have all the information required to make predictions about the behaviour of the quarter-ring TLD, tests were made for configuration C90, although it was expected that the TLD would be severely mistuned to the structure and, therefore, provide little damping.

The results of these tests can be seen in Figure 23 and Table 7.

Configuration C0 had the best results, and configuration C90 had the worst, with the formation of only one peak of displacement (at a frequency of 0.72 Hz) that was very pronounced. This was expected as the TLD in this configuration was not tuned to the structure.

The performance of the other configurations fell in between these and became progressively worse as the angle between the excitation and the centre of the TLD increased. The peaks became more pronounced, and the resonance tended towards 0.72 Hz.

As can be seen, all configurations (except C90) had the displacement peak at the lower resonance frequency being more pronounced, which can be an indicator of positive mistuning.

The attenuation of the response at the lower resonance was 71% for configuration C0 and 20% for C90. The other configurations’ attenuations varied from 68% to 64%, being a little less effective than the first configuration, as expected.

### 3.3. Tests with the Larger Pendulum

As was the case in the previous section, tests with just the pendulum were performed to confirm its natural frequency and to determine its response without vibration control.

Initially, the pendulum was excited until it reached a steady state of movement at the expected resonant frequency of 0.69 Hz. Once the excitation was stopped, the structure entered free vibration, with the displacement decaying at a rate equal to its natural frequency, in this case approximately 0.68 Hz.

Once this was determined, the sweep frequency tests were performed. Figure 24 depicts the frequency response curve of these tests based on the steady-state response of the pendulum to a base excitation amplitude of 0.5 mm.

With the natural frequency of the pendulum and its maximum steady state response determined, it was possible to calculate its damping coefficient through the decay of the free vibration until it reached 50% of the maximum response, expressed analytically in Equation (11).
(11)$$u(t)=0.0349\times e^{-0.00688\times t},$$
with u(t) being the displacement [m] and t being the time [s].

And then, with the application of Equation (6), it is possible to determine the damping coefficient of the pendulum:ξ=0.16%

A summary of the information gathered in these tests is presented in Table 8.

The results were similar to that of the smaller pendulum, with a relatively high structure displacement despite the small base displacement and an insignificant damping coefficient; however, both of these metrics were more accentuated in this case.

### Interaction of the Quarter Ring TLD with the Larger Pendulum

When the central cell of the quarter-ring TLD was aligned with the movement of the pendulum (denoted as configuration Q0; shown in Figure 25), the length of the TLD was 0.426 m. To determine the height of water to be used, the characteristics of a single cell were imputed in Equation (1), similarly to what was carried out in Section 3.3:$$h_{f}=3.5\;cm$$

With this information, the ratio of masses and the liquid depth ratio for a single cell can be determined as per Equations (2) and (3), respectively:μ=0.81%   ε=0.16

Table 9 contains the conditions in which the tests with the quarter ring TLD were performed.

For configuration Q0, due to the observed behaviour of the ring cell reservoir in Section 3.2.2 and making a comparison with the behaviour of the rectangular reservoir in Section 3.2.1 and Section 3.1, it can be assumed that the behaviour of each cell is going to be very similar in terms of sloshing frequency and liquid depth ratio, as the height of water is constant.

Therefore, Equations (4) and (5) will result in the same characteristics of a singular cell with the same height of water.
$$f_{0}=0.68\;Hz\;\;\varepsilon_{0}=0.16$$

As for configuration Q90, the centre cell will have a different sloshing frequency than the rest, which should impact the performance of the TLD. Keeping the assumption of a rectangular reservoir, with Equations (1) and (3), it is possible to calculate the sloshing frequency and the liquid depth ratio.
$$f_{f}=0.53\;Hz\;\;\;\varepsilon =0.14$$

Therefore, Equations (4) and (5) can be applied to determine the characteristics of the MTLD.
$$f_{0}=0.61\;Hz\;\;\;\varepsilon_{0}=0.15$$

In both cases, the ratio of masses is five times that of a singular cell, resulting in a value of
μ=4.05%

Figure 26 and Table 10 contain the results of these tests.

All tests showed a great attenuation of the structure displacement: 81% for configuration Q0 lower resonance and 75% in configuration Q90. These are roughly equivalent to an increase in the structural damping coefficient by a factor of five.

It is important to note that the higher resonance for configuration Q0 was barely noticeable (with a 92% attenuation). There was also a relatively high displacement (close to that of the resonance) plateau formed near the frequency 0.62 Hz. The tests performed with the rotated TLD led to the formation of a one-peaked curve, which was expected due to the mistuning in effect in these tests. 

## 4. Conclusions

The known analytical methods used for designing rectangular TLDs proved to be adequate for achieving a good performance of these vibration control systems. This study further corroborates that the liquid depth ratio is an important parameter when designing rectangular TLDs, as was pointed out by Sun [23] and Fujino et al. [24].

The water inside the reservoir seemed to follow the path of least resistance, even when this would result in a movement that was not aligned with the excitation of the reservoir, namely in the cases where the reservoir was rotated.

When dealing with the ring cell reservoir, it was demonstrated that the design methods used for rectangular TLD were also applicable, even though there was a small decrease in the performance of the TLD in those conditions. This also seems to be the case for the behaviour of the liquid inside the compartmentalised annular TLD.

The application of the MTLDs in the form of the quarter ring TLD proved to be an adequate solution for the relatively high displacement of the structure even when the reservoir had a 90° rotation.

In addition, due to the behaviour of the water inside the reservoirs when rotated, an annular TLD similar to that in the study (but without the connections between cells) could be used to control more than one mode of vibration of a chimney, as each of the rotated cells has two natural frequencies and would have significant contributions in both configurations.

Furthermore, the vibration control provided by the quarter ring was satisfactory in all studied cases. This is further corroborated as tests with this TLD were also performed with an excitation amplitude of 0.25 mm, and their results were close to identical to the ones presented.

Finally, it is worth mentioning that the use of customised sensors instead of commercial ones during the experimental tests resulted in more affordable instrumentation solutions without compromising the accuracy of the measurements.

## Figures and Tables

**Figure 1 sensors-24-02800-f001:**
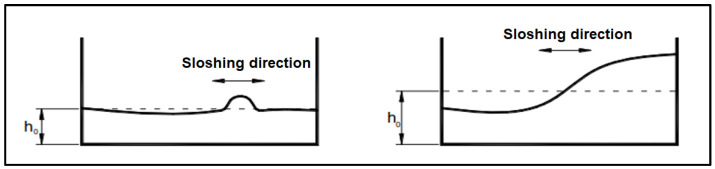
Low (**left**) and high (**right**) fill sloshing types in rectangular containers [18] (p. 6.32).

**Figure 2 sensors-24-02800-f002:**
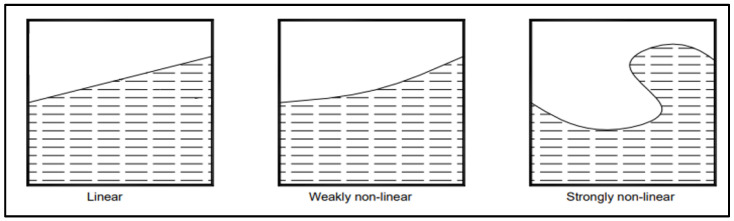
Generic liquid motion types in rectangular containers [18] (p. 6.31).

**Figure 3 sensors-24-02800-f003:**
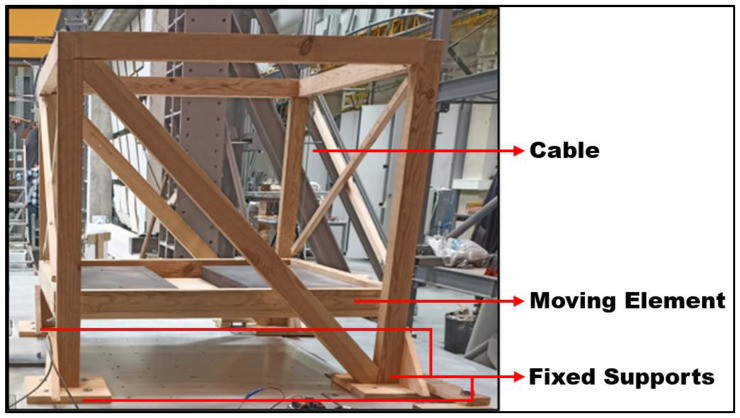
Lateral view of the smaller pendulum without any TLD fixed to the shaking table (the 7.15 kg steel plates are not shown in the picture) with its main components highlighted.

**Figure 4 sensors-24-02800-f004:**
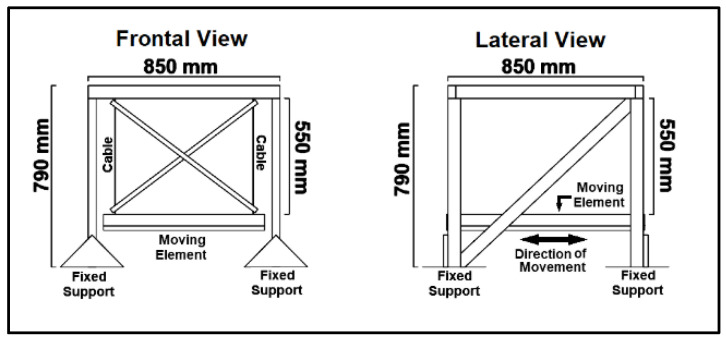
Drawings of the frontal (**right**) and lateral (**left**) views of the smaller pendulum.

**Figure 5 sensors-24-02800-f005:**
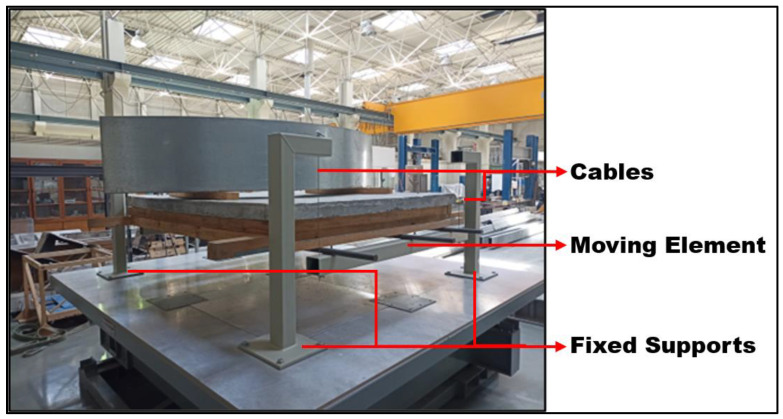
3/4 view of the larger pendulum fixed to the shaking table (the quarter-ring TLD is installed on the pendulum) with its main components highlighted.

**Figure 6 sensors-24-02800-f006:**
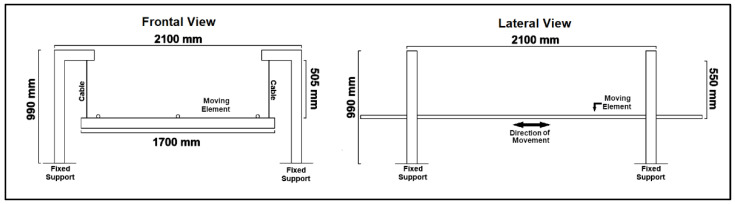
Drawings of the frontal (**right**) and lateral (**left**) views of the larger pendulum (without the “mass”).

**Figure 7 sensors-24-02800-f007:**
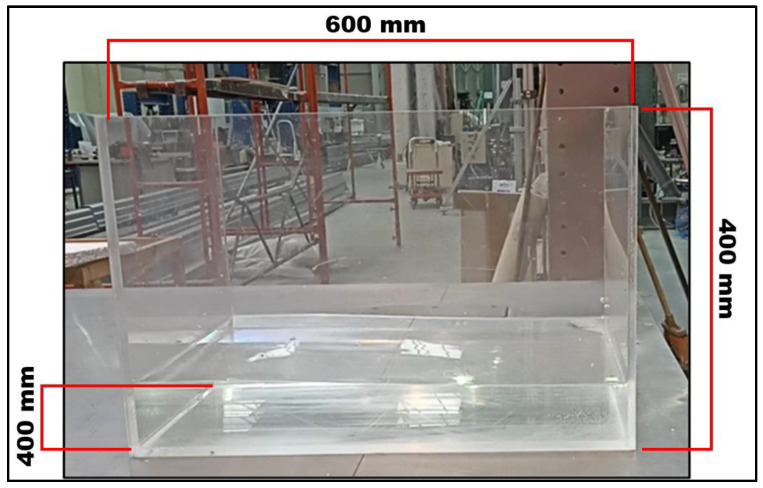
Lateral view of the rectangular reservoir filled with water (with a height of 10 cm) with its dimensions highlighted.

**Figure 8 sensors-24-02800-f008:**
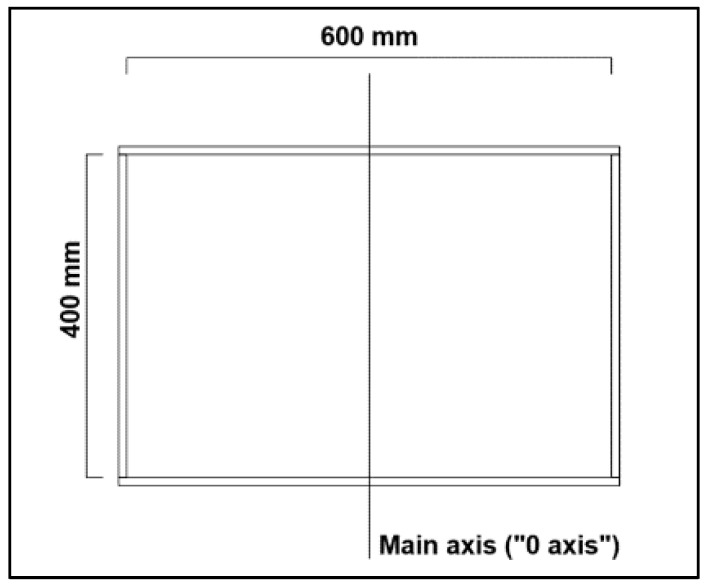
Drawing of the rectangular reservoir superior view.

**Figure 9 sensors-24-02800-f009:**
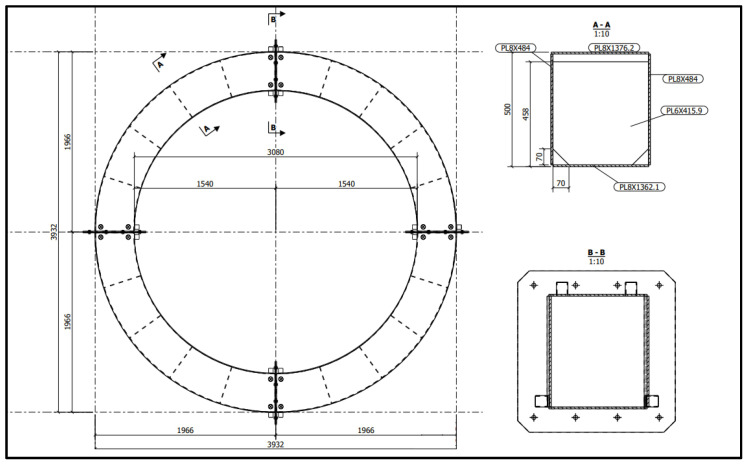
Schematics of the annular TLD (top view and two cross-sections).

**Figure 10 sensors-24-02800-f010:**
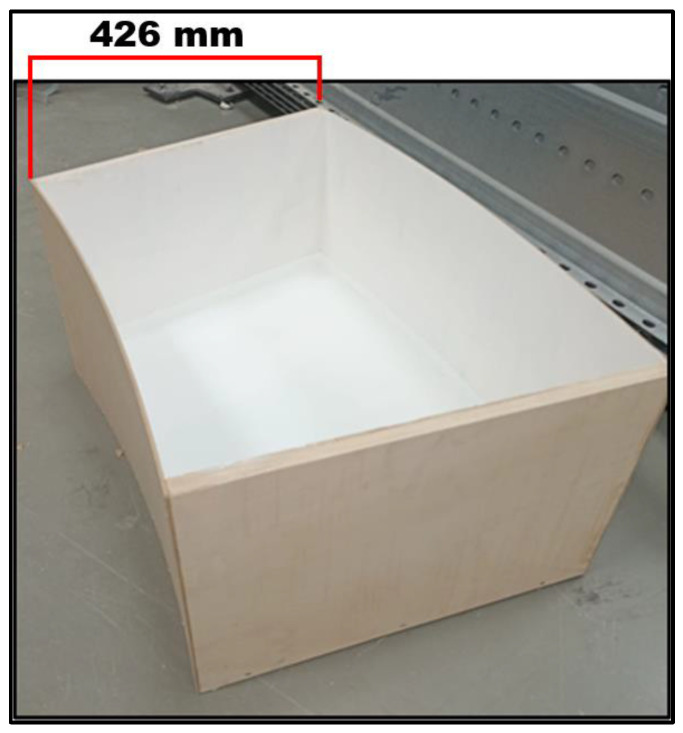
A 3/4 superior view of the cell reservoir showing its isolated interior surface, with its length highlighted.

**Figure 11 sensors-24-02800-f011:**
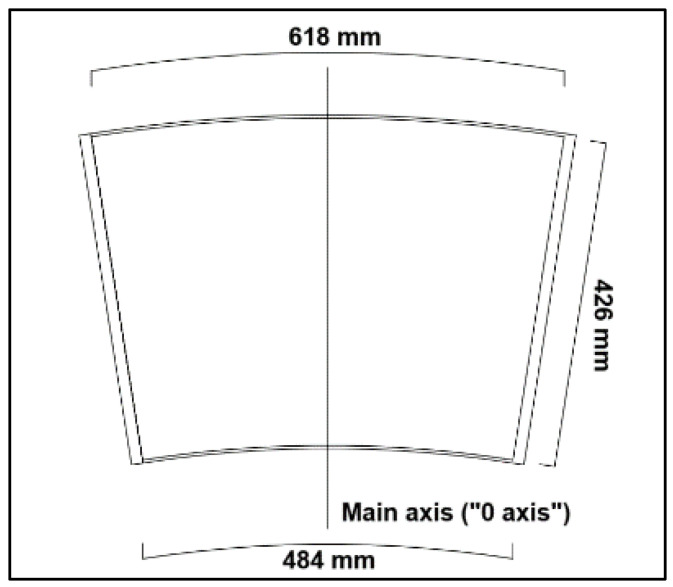
Drawing of the cell reservoir superior view.

**Figure 12 sensors-24-02800-f012:**
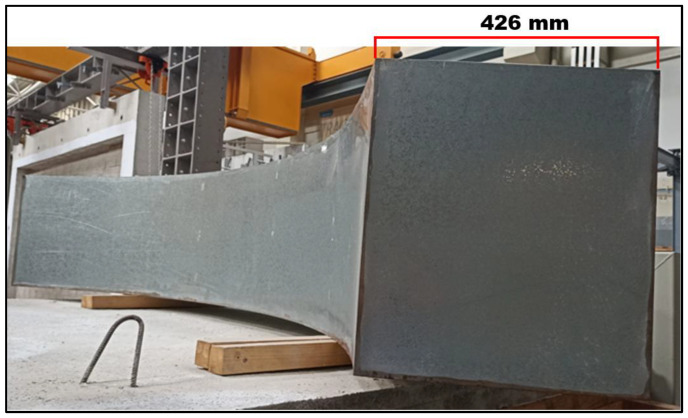
A 3/4 view of the exterior of the quarter-ring reservoir with its length highlighted.

**Figure 13 sensors-24-02800-f013:**
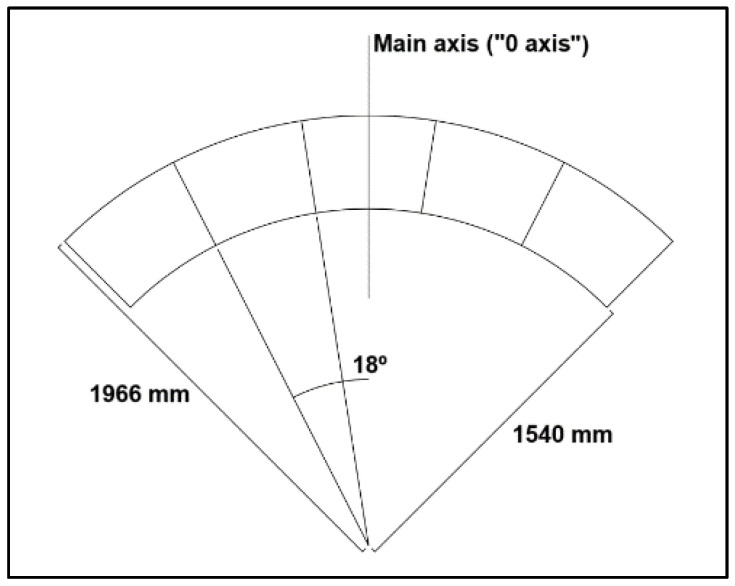
Drawing of the quarter-ring reservoir superior view.

**Figure 14 sensors-24-02800-f014:**
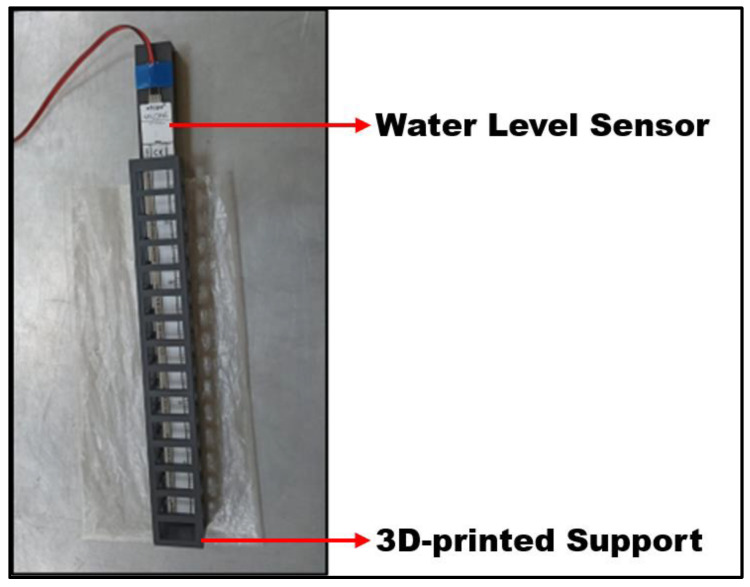
Frontal view of the water level sensor with its main components highlighted.

**Figure 15 sensors-24-02800-f015:**
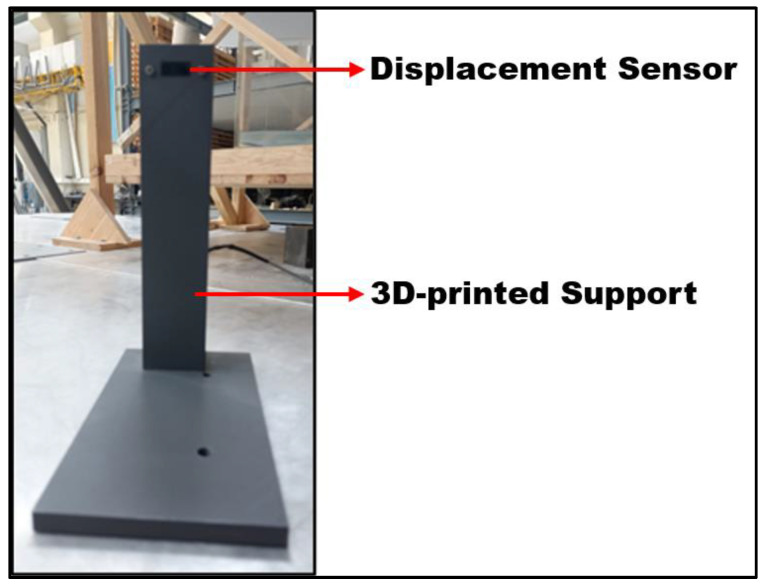
Frontal view of the displacement sensor with its main components highlighted.

**Figure 16 sensors-24-02800-f016:**
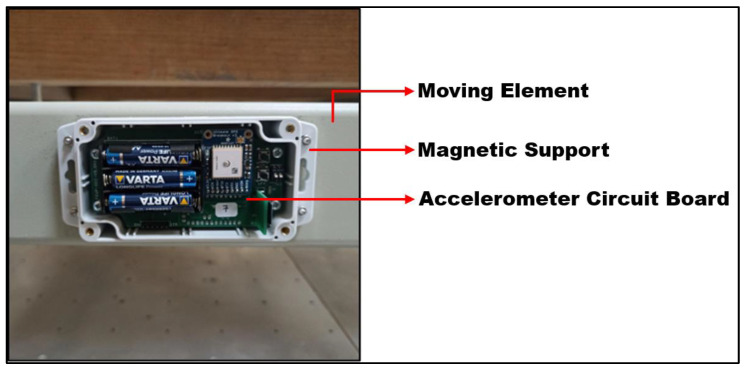
Frontal view of the accelerometer (showing its circuit board) fixed to the larger pendulum moving element by its magnetic support.

**Figure 17 sensors-24-02800-f017:**
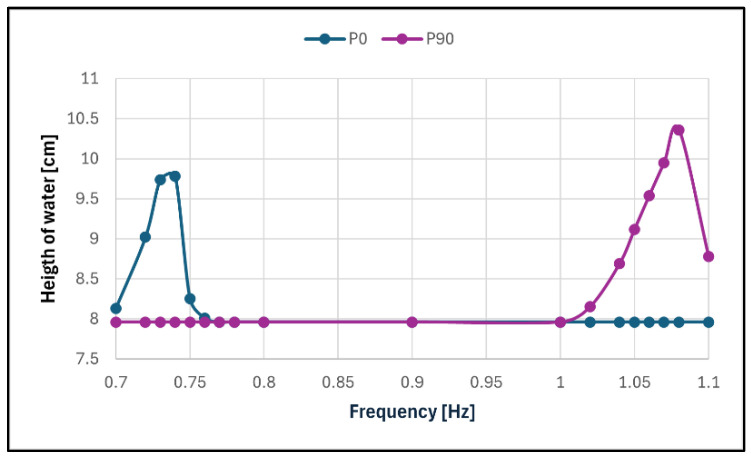
Frequency response in configurations P0 and P90.

**Figure 18 sensors-24-02800-f018:**
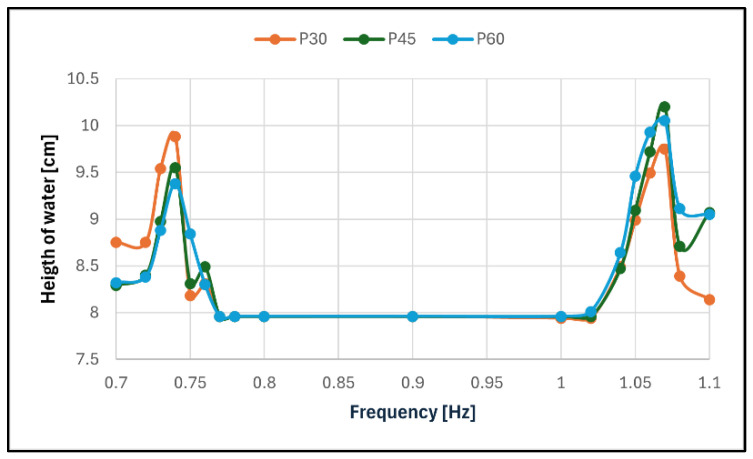
Frequency response in configurations P30, P45 and P60.

**Figure 19 sensors-24-02800-f019:**
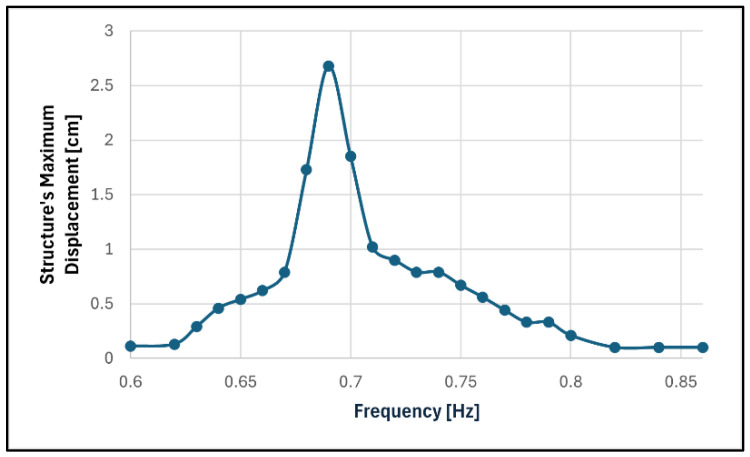
Frequency response of the smaller pendulum without TLD.

**Figure 20 sensors-24-02800-f020:**
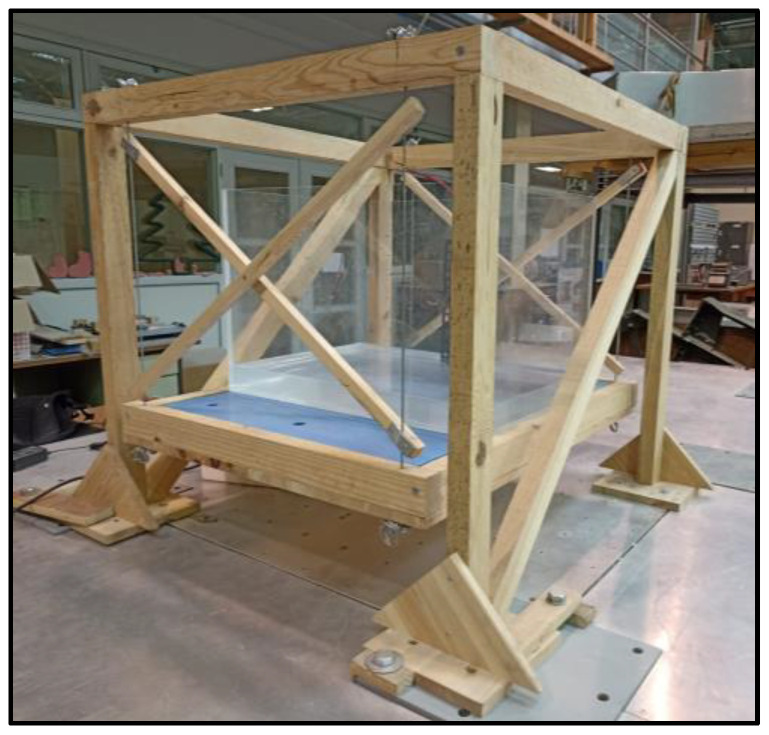
Smaller pendulum with the rectangular reservoir in configuration R0.

**Figure 21 sensors-24-02800-f021:**
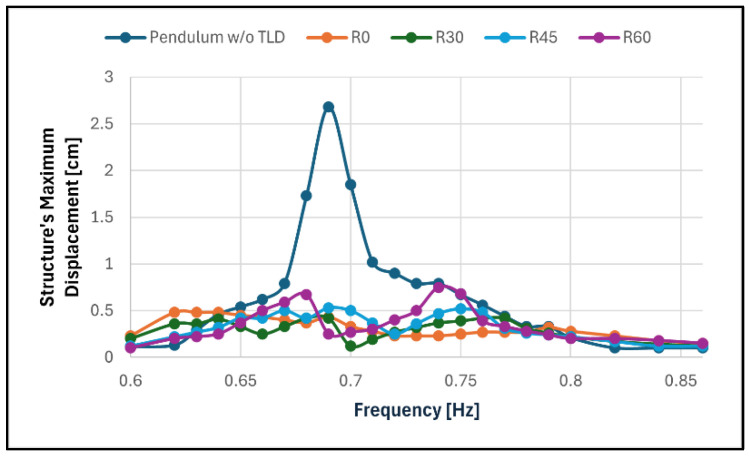
Frequency response of the smaller pendulum in configurations R0, R30, R45 and R60 when compared to its response without TLD.

**Figure 22 sensors-24-02800-f022:**
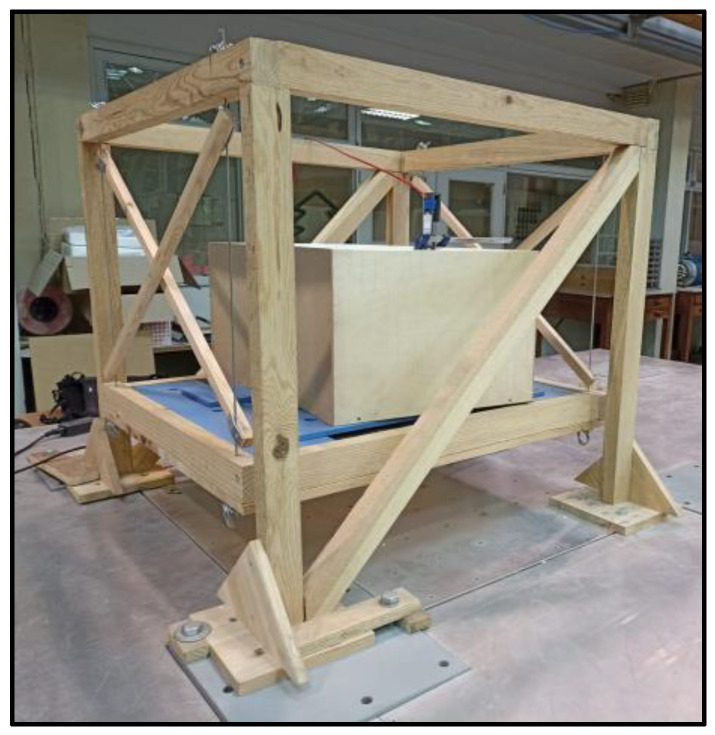
Smaller pendulum with the ring cell reservoir in configuration C0.

**Figure 23 sensors-24-02800-f023:**
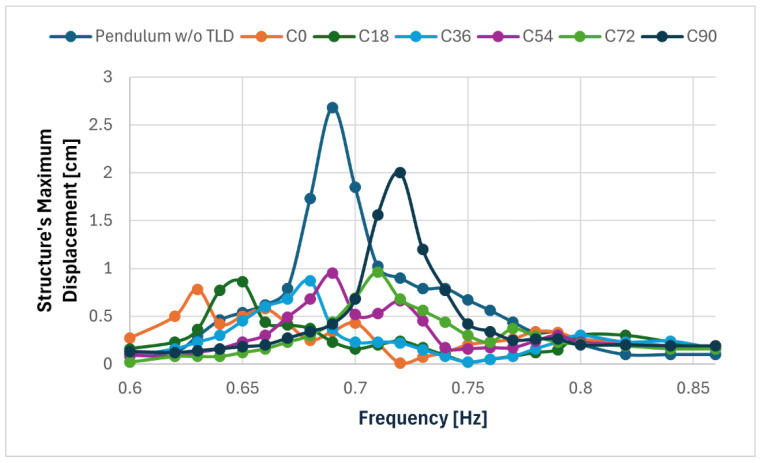
Frequency response of the smaller pendulum in configurations C0, C18, C36, C54, C72 and C90 when compared to its response without TLD.

**Figure 24 sensors-24-02800-f024:**
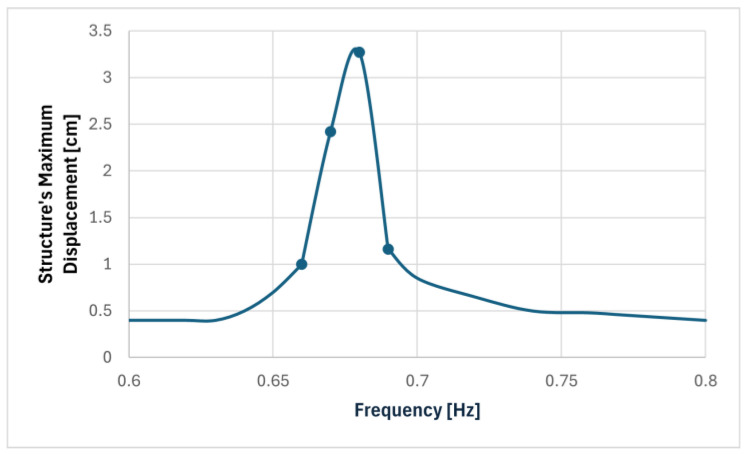
Frequency response of the larger pendulum without TLD.

**Figure 25 sensors-24-02800-f025:**
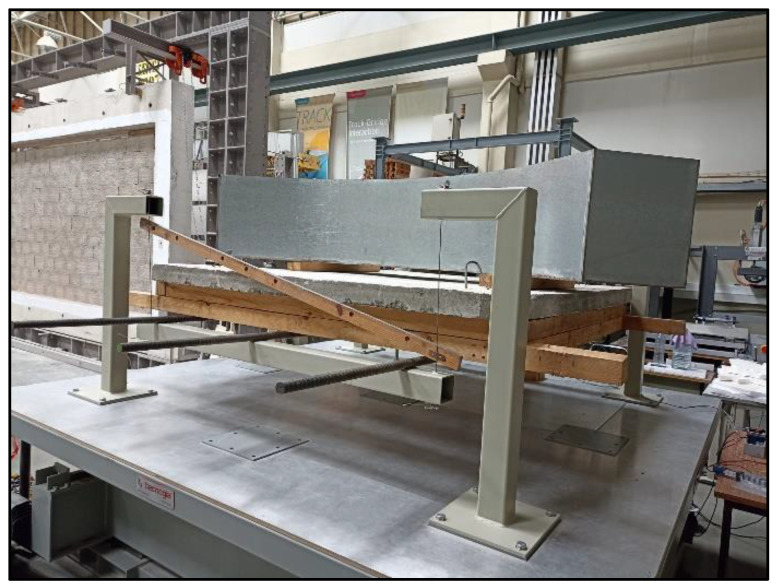
Lager pendulum with the ring cell reservoir in configurations Q0.

**Figure 26 sensors-24-02800-f026:**
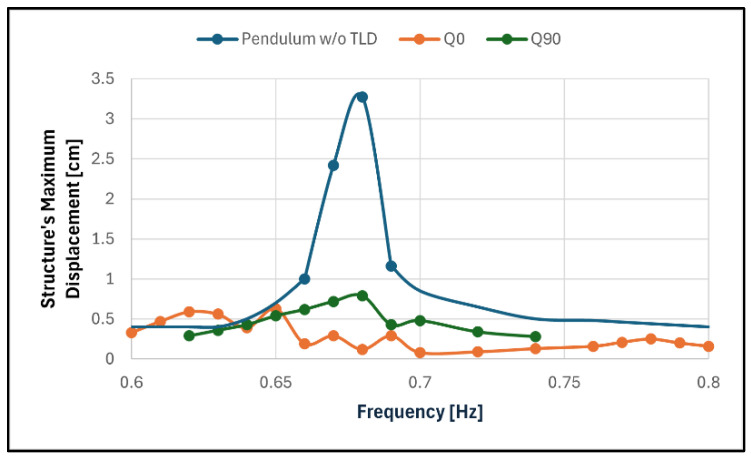
Frequency response of the larger pendulum in configurations Q0 and Q90 when compared to its response without TLD.

**Table 1 sensors-24-02800-t001:** Information about the tests with the rectangular reservoir.

Test ID	Excitation Angle to the Longer Side of the TLD	Height of Water [cm]	Excitation Amplitude [mm]	Expected Sloshing Frequency [Hz]
P0	0°	8	0.5	0.72
P30	30°	8	0.5	-
P45	45°	8	0.5	-
P60	60°	8	0.5	-
P90	90°	8	0.5	1.04

**Table 2 sensors-24-02800-t002:** Results of tests with the rectangular reservoir.

Test ID	Sloshing Frequency [Hz]	Liquid Surface Elevation [m]	Damping Ratio
P0	0.74	0.0178	0.19%
P30	0.74	0.0188	0.20%
	1.07	0.0175	0.12%
P45	0.74	0.0155	0.17%
01α90	1.07	0.0220	0.12%
P60	0.74	0.0138	0.14%
01α90	1.08	0.0205	0.12%
P90	1.08	0.0236	0.12%

**Table 3 sensors-24-02800-t003:** Results of tests with the smaller pendulum.

Oscillation Frequency [Hz]	Excitation Amplitude [mm]	Structure’s Maximum Displacement [cm]	Structure’s Damping Ratio
0.69	0.5	2.68	0.37%

**Table 4 sensors-24-02800-t004:** Information regarding the tests with the rectangular TLD on the smaller pendulum.

Test ID	Excitation Angle to the Smaller Side of the TLD	Height of Water [cm]	Excitation Amplitude [mm]
R0	0°	3.25	0.5
R30	30°	3.25	0.5
R45	45°	3.25	0.5
R60	60°	3.25	0.5

**Table 5 sensors-24-02800-t005:** Results of tests with the rectangular TLD on the smaller pendulum.

Test ID	Resonant Frequency [Hz]	Maximum Structure’s Displacement [cm]	Attenuation of the Structure Response
R0	0.63	0.48	82%
	0.79	0.32	88%
R30	0.64	0.41	85%
	0.76	0.42	84%
R45	0.67	0.50	81%
	0.75	0.52	81%
R60	0.68	0.67	75%
	0.74	0.75	72%

**Table 6 sensors-24-02800-t006:** Information about the tests with the ring cell TLD and the smaller pendulum.

Test ID	Excitation Angle to the Centre of the Ring Cell TLD	Height of Water [cm]	Excitation Amplitude [mm]
C0	0°	3.7	0.5
C18	18°	3.7	0.5
C36	36°	3.7	0.5
C54	54°	3.7	0.5
C72	72°	3.7	0.5
C90	90°	3.7	0.5

**Table 7 sensors-24-02800-t007:** Results of tests with the ring cell TLD and the smaller pendulum.

Test ID	Resonant Frequency [Hz]	Structure’s Displacement [cm]	Attenuation of the Structure Response
C0	0.63	0.78	71%
	0.79	0.34	87%
C18	0.65	0.86	68%
	0.83	0.30	89%
C36	0.68	0.87	68%
	0.82	0.30	89%
C54	0.69	0.95	65%
	0.80	0.30	89%
C72	0.71	0.96	64%
	0.78	0.37	86%
C90	0.72	2.00	25%

**Table 8 sensors-24-02800-t008:** Results of tests with the larger pendulum.

Oscillation Frequency [Hz]	Excitation Amplitude [mm]	Structure’s Maximum Displacement [cm]	Structure’s Damping Ratio
0.68	0.5	3.27	0.16%

**Table 9 sensors-24-02800-t009:** Information about the tests with the quarter ring TLD and the larger pendulum.

Test ID	Excitation Angle to the Centre of the Quarter Ring TLD	Height of Water [cm]	Excitation Amplitude [mm]
Q0	0°	3.5	0.5
Q90	90°	3.5	0.5

**Table 10 sensors-24-02800-t010:** Results of tests with the quarter ring TLD and the larger pendulum.

Test ID	Resonant Frequency [Hz]	Structure’s Displacement [cm]	Attenuation of the Structure Response
Q0	0.65	0.63	81%
	0.78	0.25	92%
Q90	0.68	0.79	75%

## Data Availability

The data presented in this study are available on request from the corresponding author.

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
