# Peer review of "Experimental Testing on Tuned Liquid Dampers for Implementation in Industrial Chimneys"

_sensors, 2024, doi:10.3390/s24092800_

Round 1
Reviewer 1 Report
Comments and Suggestions for Authors
The manuscript “Experimental testing on Tuned Liquid Dampers aiming at their implementation on industrial chimneys” studies the damping characteristics of water-filled reservoirs and their effect on the oscillations of structures.
Despite the interesting experiment the authors limit their attention to a narrow field of interest. Only one kind of liquid is considered, moreover water is not the most effective damping medium. Then only one amplitude is used, although the authors pay attention to non-linear effects (Figure 2). The experiments confirmed effective damping due to water, but these results cannot be generalized because of limited data.
I would like to recommend revising of the paper with the goal to present more useful findings.
Please, take into account following remarks as well:
1. The content of the manuscript does not include any ideas for TLD implementation in industrial chimney. The title should be revised.
2. Direct citations (lines 29-31, 37-41, 276-279…) are not appropriate.
3. The definition of fmax is not clear. Strictly speaking, the highest natural frequency is not limited.
4. The relation between a mechanic pendulum and studied construction is not obvious. It should be very carefully proved. What is the length Lp in Eq.(9) in relation to the pendulum shown in Figures 3 and 4?
5. The drawings of both pendulums with all dimensions should be included.
6. The drawings of three reservoir with all dimensions should be provided. Please, show the direction of the zero excitation angle in this drawing.
7. Figures 8, 10,11 are not informative and can be omitted.
8. Technical details like in lines 244-246, 264-267, etc. should be omitted.
9. Lines 415-420. It would be interesting and important to include photos or sketches of disturbed water surfaces to compare them with Figure 2?
10. Figure 19 is very bad; it does not prove the resonant oscillation of the larger pendulum. The displacement should be plotted over a wider frequency range.
Author Response
Dear Reviewer,
Firstly, we would like to thank you for the review and your insight on what should be improved in this study.
1) As you pointed out we only used water as the liquid even though it is not the most effective damping medium, this was because the study was performed under request of Taylor Devices Europe scrl and they request that we did the tests with water specifically (as it is the liquid they use in their TLD).
2) We would also like to inform that we performed tests with the quarter-ring TLD with another amplitude and the results were pretty much identical to those presented, we choose to not include them as they did not bring more information. We will, however, specify this in the text for clarity.
- We agree that we did not address their implementation as we thought that this was clear as the TLD was developed with the intent of being applied to chimneys. More information will be included in the text to correct this mistake.
- The direct citations were used to facilitate the consultation of the reference documents and to preserve the intent of their authors. If it is not mandatory, we would prefer to maintain the direct citations.
- The “fmax” was, in this case, the sloshing frequency of the TLD with the highest natural frequency. The name of the variable was changed to clarify this.
- The relation between the pendulum and the chimney is due to both having the same natural frequencies and amplitudes of displacement when in resonance, and negligible damping coefficients. This was reinforced in the text.
- 4/5/6. The length of the pendulum was determined as the free length between the base of the support structure and the top of the mobile part (the pendulum itself). Drawings were included in the text to address these points.
- Although these figures are not informative, as this study is meant to be published in the special issue related to “Novel Sensors for Structural Health Monitoring”, we believe that they should be included in the article.
- We agree that this information is unnecessary, thus it was omitted.
- We agree that this would be interesting indeed. However, we tried to do this when first developing the document, but the imagens taken had not the enough quality and clarity to be included in the paper, nor to be used as a reference for drawings.
- We plotted the wider frequency range using the theoretical curve as a basis to better transmit how the structure would behave in other frequencies (as there is not experimental data for those frequencies); but, as the pendulum is a single degree of freedom structure, finding a peak of displacement (especially one that is close to that obtained with the analytical formula and that matches the frequency of free decay) is enough to prove that it is the resonance.
Thank you,
The authors

Reviewer 2 Report
Comments and Suggestions for Authors
This is a potentially interesting submission. The experiments presented may always bring new interesting knowledge. However in this case the literature review is not properly carried.
lines 144 - 147, Quote:
(...) experimental tests with non-rectangular TLD are still uncommon and tests with a TLD of this particular configuration is almost unheard of. The equipment used in this study is also a notable aspect, as TLD tests carried out on large shaking tables are also uncommon.(...)
This is not true. Simply check the literature (rather old in some cases) and make a thorough review of how TLD were tested on shaking tables and declare what new is proposed to be checked in this experimental campaign.
Add two drawings that will explain how each of the pendulum of both experiments are constructed. Show details with arrows and text explanations. Add a separate chapter describing in detail how the shaking table was excited. Which of the 3 table directions were active? What seismic time history was applied (in this case show time history records and their Fourier spectra). Or/and perhaps sinusoidal excitations were carried out. All this must be clearly explained. All this is unclear in this submission.
The abstract seems to contain more information on the results than conclusions which are rather trivial. Such experiments should bring specific, novel results. Writing that some results of previous studies (which ones?) were confirmed is not enough.
Comments on the Quality of English LanguageOnly minor problems occur. E.g. some typo errors e.g.:
line 132 laborital (???)
Author Response
Dear Reviewer,
Firstly, we would like to thank you for the review and your insight on what should be improved in this study.
Additional literature used in the development of the study was provided. As for the lines 144-147, we agree that we made a mistake in the writing, leading to a misleading statement, this was corrected to better express what we originally meant with this sentence.
Drawings were included in the text to address this point and it was clarified that the excitation in use was sinusoidal. We think that an additional chapter describing the detailed functioning of the shaking table would draw reader's attention to a topic that is out of the scope of this research.
We agree with the last comment, therefore we improved the conclusion chapter of the article.
Additionally, we made an additional revision of the document aiming at correcting all the typos and grammatical mistakes (including the one mentioned).
Thank you,
The authors

Round 2
Reviewer 1 Report
Comments and Suggestions for Authors
Dear Authors,
Thank you for the revision. I found that the experimental results are interesting enough to be published, whereas the style of the paper must be improved (direct citations are not OK4, speculative relation between the pendulum and studied structure, non-informative pictures like Figure 16, etc. should be revised)
Author Response
Dear Reviewer,
Once more, we would like to thank you for the review and your insight.
As required, the direct citations were replaced with text.
The reasons behind the pendulum being used as a stand-in for the structure were further clarified. Actually, it is very difficult to construct a 1-DOF system with low frequency and low damping using a mass sliding in bearings and connected to springs (due to friction problems). Therefore, the best way of constructing a 1-DOF system with similar characteristics is to use a pendulum. Note that only 1-DOF is being considered, corresponding to the 1st vibration mode of the chimney which is the critical one.
As for the figures, as we expressed before, despite not being informative, we would like to keep them on the paper because they were the only images of the custom sensors developed for this study. It would be disappointing if images of the sensors were not present in the article, as the special issue of this edition is related to this topic.
Thank you,
The authors

Reviewer 2 Report
Comments and Suggestions for Authors
The Authors partly improved their narration by adding e.g. some figures, though without explanation notes. While Figure 6 explains itself yet Figure 4 requires more explanations. Perhaps an image with dashed lines of a tilted pendulum could be over-imposed in the figure? Still, some arrows with explanations added to all additional figures could be helpful for the reader of "Sensors".
Author Response
Dear Reviewer,
Once more, we would like to thank you for the review and your insight.
The figures in question were modified to show the direction of movement and to better explain which part is which (fixed supports, moving element, direction of movement). We also specified what drawing corresponds to each view (frontal and lateral).
Thank you,
The authors
